# Fonio and Bambara Groundnut Value Chains in Mali: Issues, Needs, and Opportunities for Their Sustainable Promotion

**Charlie Mbosso [1], Basile Boulay [1], Stefano Padulosi [1,*], Gennifer Meldrum [1], Youssoufa Mohamadou [2], Aminata Berthe Niang [2], Harouna Coulibaly [2], Yara Koreissi [2] and Amadou Sidibé [2]**

1    Bioversity International, Via dei Tre Denari, 472/a, Maccarese, 00054 Rome, Italy; mbossopc@gmail.com (C.M.); basileboulay@gmail.com (B.B.); G.Meldrum@cgiar.org (G.M.)
2    Institut d'Economie Rurale, Rue Mohamed V Bamako, BP 258 Bamako, Mali; YoussoufaMohamadou@yahoo.fr (Y.M.); aminataberthe@gmail.com (A.B.N.); hscoulibaly@yahoo.fr (H.C.); ykoreissidemb@gmail.com (Y.K.); amadousidibe57@yahoo.fr (A.S.)
*    Correspondence: s.padulosi@cgiar.org

**Abstract:** As the effects of climate change are severely straining West African agricultural systems, the adoption of more incisive interventions in support of sustainable development agendas for the region is highly critical and cannot be further delayed by governments. Neglected and underutilized species (NUS) are one important ally in pursuing resilience in both production and food systems because of their promising traits in terms of nutrition, adaptation to local agroecosystems, and economic potential for local populations. Focusing on fonio, a gluten-free traditional cereal, and Bambara groundnut, a protein-rich leguminous crop, we investigate issues in their production, commercialization, and consumption in southern Mali. The aim was to assess needs and opportunities for improving their value chains and increasing their use and societal benefits. Using a Rapid Market Appraisal method, we surveyed traders, producers, processors, and consumers of target crops in 2017 and 2018. Our findings indicate that while both crops are consumed and praised by local populations, critical bottlenecks inhibit their wider socioeconomic potential. Lack of access to inputs and equipment and presence of sand in the commercialized product are important issues for fonio, whereas the Bambara groundnut value chain suffers from poor processing facilities and lack of market promotion. Policy recommendations to tackle the identified bottlenecks are proposed.

**Keywords:** Mali; sustainable agriculture; fonio; Bambara groundnut; underutilized crops

## 1. Introduction

Research on the effect of climate change in agriculture in West Africa shows that crop mixes will need to adapt in the coming years, as the effects will have a direct impact on livelihoods. A study by Traore et al. [1] shows that cotton growing in the Sikasso region is likely to be adversely affected by rising temperatures in Mali and changes in rainfall patterns. Looking at sorghum and millet in West Africa, Sultan et al. [2] evaluated the effects of 35 possible climate scenarios on yields and found negative impacts in 31 cases. In this context, rural populations reliant on rain-fed agriculture are likely to be hit the hardest [3]. In Mali, the agricultural sector will experience continued soil fertility depletion and water stress, especially in dry areas [4]. Production challenges faced by smallholder farmers in Mali are likely to be heightened as the effects of climate change amplify in the region, possibly with an additional burden in terms of increased malnutrition and reduced food security [5]. The former could

severely increase and disproportionately affect children, together with rising anemia and stunting rates [6].

In this context, the current research focus on a few main cereal crops in agricultural policies seems misguided. On the other hand, marginal crops can become a building block of future strategies while also contributing to the Sustainable Development Goals [7]. From an agronomic perspective, yields of major crops are expected to decrease in West-African countries, no matter the climatic scenario considered [8]. A shift of attention away from major crops and towards lesser-known and documented minor crops could prove a key ingredient in the development of sustainable agricultural agendas [9,10]. Minor crops deserve attention for at least four reasons: (i) They tend to be very nutritious, (ii) they require few inputs and are often indigenous to African regions, (iii) they maintain unique genetic diversity, and (iv) they contribute to household income [7]. Furthermore, as women are mostly responsible for the farming and/or processing of subsistence crops in the African context, improved utilization of minor crops could enhance women's access to supply chains and income-earning opportunities [11], as well as improving nutritional outcomes at the household level [12]. In the literature, minor crops are often referred to as underutilized, marginal, and 'neglected and underutilized species' (NUS).

In this study, we focus on two such crops grown in Mali: Fonio (*Digitaria exilis* (Kippist) Stapf) and Bambara groundnut (*Vigna subterranea* L. Verdc.). Fonio is one of the most ancient indigenous West African cereals and is a major part of the diet in some communities in Mali [13–15]. It is an excellent source of protein that is rich in the sulfur-containing amino acids methionine and cysteine, which are deficient in rice, maize, and sorghum [16], and their concentrations are slightly higher than those defined for the Food and Agriculture Organization (FAO) protein reference [17]. It is also rich in micronutrients, including iron and zinc, and therefore has the potential to improve intakes of these nutrients in deficient populations. The food composition table of Mali indicates fonio as the cereal with the second-highest iron content (8.5 mg/100 g dry matter) after sorghum (11 mg/100 g dry matter) [18]. Furthermore, fonio has a low glycemic index, and it may serve as an alternative grain for people with gluten intolerance [16]. Bambara groundnut is a legume crop native to Africa, commonly grown for its seeds by subsistence farmers [19]. In certain regions, its importance as a leguminous crop is only matched by groundnut and cowpea [20]. Bambara groundnut is rich in proteins, with content varying between 14 and 24 g per 100 g, and is also rich in carbohydrates, with 28 to 40 g per 100 g [21]. The crop also provides fatty acids and minerals: A 100 g portion serving of Bambara groundnut fulfils more than half the recommended daily allowance for potassium intake for children and adults and covers the entire recommended daily allowance for magnesium and zinc [22]. Praised for its agronomic properties, Bambara groundnut is relatively drought-tolerant and requires minimal chemical inputs. Furthermore, it acts as a natural nitrogen fixer and can therefore enhance the yields of non-nitrogen-fixing crops when properly intercropped [23]. Bambara groundnut cultivation relies on landraces, which are highly suited to local agro-ecologies [24].

Despite their excellent nutritional profiles, hardiness and versatility in use, the cultivation and trade of both fonio and Bambara groundnut remain below their potential throughout sub-Saharan Africa. The situation is slowly changing for fonio, however. Regarded for a long time as a minor crop without economic potential [25], it is attracting renewed interest in urban areas of West Africa for its organoleptic and nutritional qualities [26,27]. Recently, the crop has been listed as a priority crop for West Africa [28]. The pace of change is slower for Bambara groundnut, despite efforts at increasing the visibility of the crop over the past years, such as being branded as one of the FAO's "traditional crops". Both these crops could help address key challenges faced by the Malian agricultural sector. For example, the nitrogen-fixing properties of Bambara groundnut can help maintain soil fertility, and its low water requirements can be an advantage in dry or drought-prone areas. Fonio thrives in the semi-arid lands of the Sahel thanks to its low water requirements, an extensive root system that helps the plant to draw water from deep underground, and its fast maturation. Fonio has long been considered a strategic food for rural West Africans, being the first crop to be harvested in the "hungry

season", which is a time of critical shortage before other staples, like sorghum or maize, are ready for harvest [29]. It is able to grow in poor soils without the use of fertilizers and, hence, is typically planted later in crop rotation cycles, after maize or sorghum.

Among the factors explaining why fonio and Bambara groundnut remain underutilized, weak value chains are a key issue. For fonio, several studies point towards the lack of appropriate technology for harvesting, threshing, and processing [15,30]. For Bambara groundnut, trade is often confined to the immediate village boundaries, with little commercialization or processing involved [31,32], and negative traditional beliefs are associated with the crop in parts of Africa (e.g., [33] for Malawi). However, few studies have specifically examined issues of marketing for Bambara groundnut [19,34,35] and little information is available on its value chain in Mali. Similarly, for fonio, most of the studies looking at Mali only focus on specific actors along the value chain in a specific location (e.g., [15] or [36] for urban consumers of fonio in Bamako). In this study, we aimed to broaden understanding of factors limiting the use of these crops in Mali by assessing barriers along their value chains.

Using a Rapid Market Appraisal method, data were collected from producers, traders, and consumers in 2017 and 2018 in three areas of southern Mali. Focus groups and individual interviews were conducted to identify the obstacles faced by actors in rural and urban areas. By focusing on several levels of the value chains, we identify the most salient constraints to greater production and commercialization of these crops and propose policy recommendations that help in mitigating them. We paid attention in order to capture the gender dimension involved in the farming, processing, and selling of the two target crops to assess opportunities for value chain development to contribute to women's empowerment. Our results show that the value chains for both crops suffer from important bottlenecks, even though the chain for fonio is more commercially developed than for Bambara groundnut, and the value chain in the case of fonio includes exports, mainly to Senegal and France, albeit in very low volumes. For the former, weak demand, presence of sand in the final products, and lack of capital for processing and marketing are key issues. For the latter, weak demand, lack of processing units, and lack of promotion prevail.

## 2. Material and Methods

### 2.1. Study Sites

As for most sub-Saharan African countries, the economy of Mali crucially relies on the agricultural sector. After a decline observed in the mid-1990s, the contribution of the agriculture, forestry, and fishing sector to GDP started to increase again from the mid-2000s, reaching 38.5% of GDP in 2018, and employing an estimated 65% of the population according to the World Bank's World Development Indicators. In terms of agricultural profile, the main food crops produced in Mali are cereal crops, especially rice, sorghum, millet, and maize [37]. The production of fonio, also a cereal, is negligible at the national scale when compared to these major crops. While the total estimated production of maize in Mali in 2015 was 2,276,000 tons, the estimated fonio production barely reached 20,000 tons. Similarly, in the case of Bambara groundnut, its production was much less than that of other competing leguminous crops, such as groundnuts. In 2015, the total estimated production of the former amounted to less than 7% of the latter, at 27,691 tons for Bambara groundnut against 421,924 for groundnut [38]. The Malian agricultural sector remains fragile and subject to many threats, the most important of which is climate change, at least in the medium to long term. Poor technological development, lack of irrigation schemes/facilities, and poor storage capacity have been identified as key constraints to the sustainable development of the sector, to which more macroeconomic aspects, such as price and exchange rate volatility, need to be further added [39,40].

This study focused on three areas in Mali that were selected to represent different situations for the use and commercialization of fonio and Bambara groundnut (Figure 1).

1.　Bamako: Situated in the central south of Mali, Bamako is the capital city, home to approximately 10% of the national population (pop. 2,009,109, Population Census 2009). Bamako is the biggest

urban center in Mali and, therefore, constituted an important 'barometer' to assess the commercial integration of marginal crops.

2.  Cercle de San and Tominian in the Ségou region, with San as a main urban center (pop. 68,078, Population Census 2009). Agriculture is the predominant sector in the Ségou region. The study area was within the north Sudano-Sahelian production zone (sensu [39]), where cropping systems are based on pearl millet and sorghum complemented by peanut. Among the different regions of Mali, the greatest production of fonio and Bambara groundnut has been recorded in Ségou, which accounted for 52% and 50% of national production of these crops, respectively, in 2015 [38]. The area around San and Tominian was targeted by the study because it is known as a major production area for fonio [40].

3.  Cercle de Koutiala and Cercle de Sikasso in the Sikasso region, with Koutiala and Sikasso as the main urban centers (respective pop. 137,919 and 225,753, Population Census 2009). Among the regions of Mali, Sikasso is an important contributor to the national agricultural output, and it benefits from above-average soil fertility. The region is the biggest cereal producer of the country, often considered Mali's 'granary'. In 2015, it produced 29% of the national cereal production and 67% of maize production [38]. The cropping systems in these areas are dominated by rainfed cotton and maize grown with mineral fertilizer and organic manure, while tubers, vegetables, and fruits are produced in lowland areas [39,41]. The Sikasso region accounted for 8% of fonio and 7% of Bambara groundnut production in Mali in 2015 [38]. However, in the focal areas of this study, traditional grain crops have been largely displaced with the expansion of cotton and maize production [42–44].

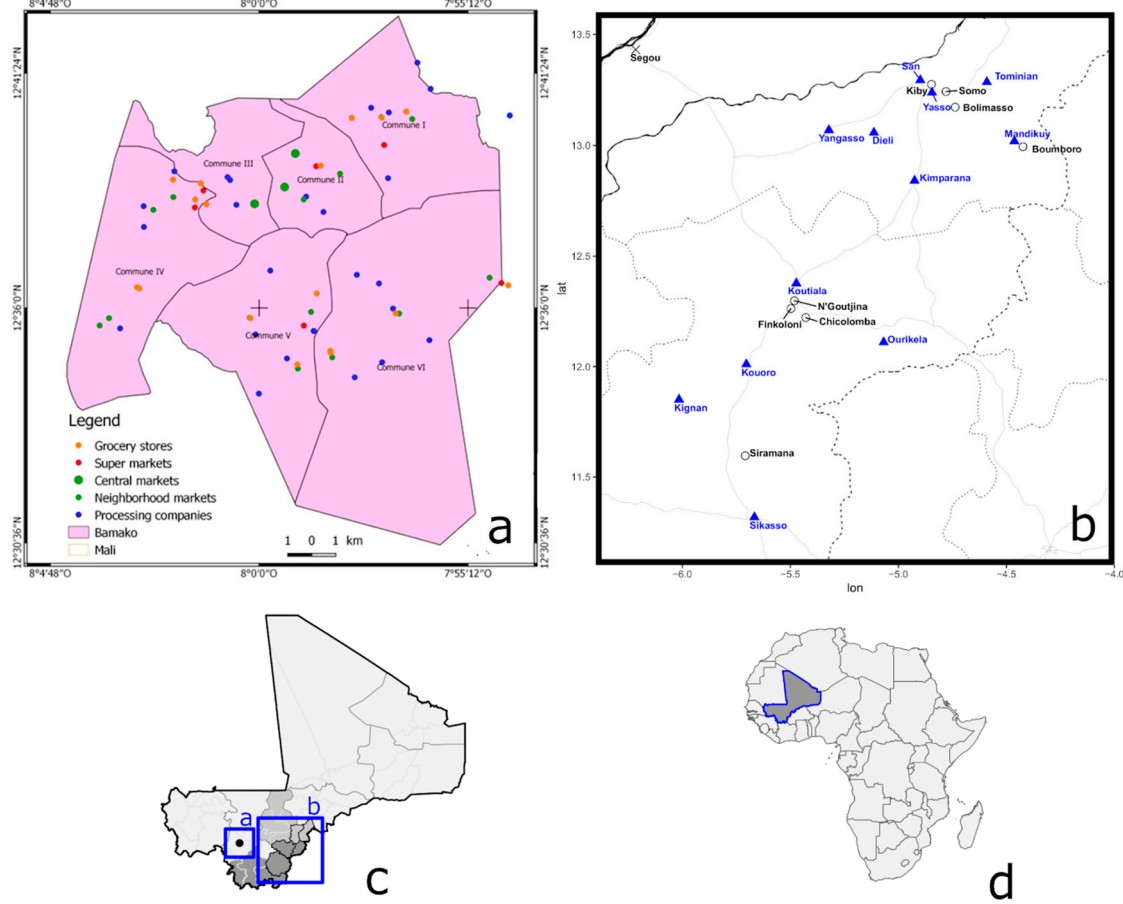

**Figure 1.** Location of study sites: (**a**) Map of surveyed markets and retailers in Bamako; (**b**) map of surveyed markets (blue triangles) and villages (open circles) in the Sikasso and Ségou regions; (**c**) location of study areas within Mali; (**d**) location of Mali within the continent of Africa.

## 2.2. Rapid Market Assessment

A rapid value chain assessment of fonio and Bambara groundnut was conducted in the study areas through semi-structured interviews and focus groups with traders, producers, and consumers complemented by direct observations. The overall aim was to understand the current state of marketing and bottlenecks in the value chains. Fonio and Bambara groundnut have different agronomic, socio-cultural, and economic dynamics. For each actor along the value chain (trader, producer, consumer), we sought to elicit key information regarding the current level of use of target crops, as well as perceptions on factors hindering their wider popularization. Topics explored in these interviews are listed in Table 1. Interview guidelines are provided in Supplementary File 1. All subjects gave their informed consent for inclusion before they participated in the study.

**Table 1.** Topics explored in surveys at trader, producer, and consumer levels during the rapid market assessment.

| Level | Topics Explored |
|---|---|
| Trader level | • Crop profile from a commercial point of view<br>• Processing methods<br>• Different forms and specificities under which the crop is sold<br>• Strengths and weaknesses of trading the products |
| Producer level | • General perception of the crop among producers<br>• Production trends, gender dynamics in production, genetic diversity<br>• Strengths and weaknesses of producing the crop |
| Consumer level | • General perception of the crop among consumers<br>• Patterns of consumption and preparation, including cultural significance of the crop in the diet<br>• Perceived benefits and drawbacks of consumption |

### 2.2.1. Trader Surveys

Nineteen markets in the three focal areas (6 in Bamako, 8 in Ségou, and 5 in Sikasso) were visited in May 2017 to assess the status of fonio and Bambara groundnut marketing and to interview traders of both crops. In the Ségou and Sikasso regions, the markets visited were located in the focal district centers (San and Tominian in Ségou; Koutiala and Sikasso in the Sikasso region) and in more rural areas. In Bamako, the markets were selected with the advice of *l'Observatoire Du Marché Agricole* (OMA) to reflect the diversity of neighborhoods found in Bamako. At least one of the focal crops was being marketed in each of the selected markets.

Convenience sampling was followed in the surveys, aiming to cover 3-5 traders of specific products in each market. The final sample depended on traders' presence, willingness, and availability to participate. The market surveys were led by two female researchers from Bioversity International and *Institut d'Economie Rurale* (IER), who were complemented by male workers from local non-governmental organizations (NGOs), who helped with surveying and translating questions in local languages. Upon arrival in the markets, the survey team identified fonio and Bambara groundnut traders, and a selection of these actors were invited to take part in semi-structured interviews. In a few markets, the team was assisted by the head of local traders, who directed the surveyors to traders dealing with the target crops. Overall, 83 fonio traders and 45 Bambara groundnut traders were interviewed concerning five fonio products and three Bambara groundnut products, which were identified to be the primary products marketed for these crops. The frequency and characteristics of the surveyed traders roughly reflected upon the traders' and products' presence in the markets. However, because the sampling was convenience-based, it cannot be considered a strictly representative sample.

A complementary survey of markets in Bamako was carried out in 2018. The aim was to explore in depth the state of fonio and Bambara groundnut marketing in different types of outlets in the six administrative communes of the city. Each commune encapsulates several neighborhoods. Five types of outlets that represented the variety of commercial structures in the capital city were targeted [45]:

(1) Supermarkets, which are modern and commercial outlets selling fresh and processed food items; (2) grocery stores, which are small self-service stores; (3) neighborhood markets, typically small open-air outlets in which individual sellers supply fresh produce to the local population; (4) central markets, which tend to supply individual traders from neighborhood markets; and (5) processing units, which sell their products largely to market retailers and restaurants, and are entirely run and operated by women. A male researcher from IER visited at least one supermarket and 3–6 of each other type of outlet in the six communes of Bamako in 2018. For the supermarkets, grocery stores, and processing units, we typically interviewed one person. In the central and neighborhood markets, we aimed to survey 3-5 traders for specific products. In total, 254 traders were interviewed from 6 grocery stores, 7 supermarkets, 16 neighborhood markets, 3 central markets, and 31 processing companies.

### 2.2.2. Producer Surveys

Fonio and Bambara groundnut producers were surveyed in eight villages in 2017, four in the Ségou region and four in the Sikasso region. The villages selected for the survey had at least a few households involved in production and sale of the focal crops. The willingness of community members to participate in the study was an important criterion guiding the selection of villages. The team selected villages that held good rapport and trust with the researchers' affiliated organizations.

Safety and accessibility by road were additional factors that guided site selection. Three of the four villages in each region were part of a project promoting value chain development of fonio and Bambara groundnut. This study was part of the initial investigations to gain insight into the existing status and constraints for using these crops, and the findings were intended to inform the development of project interventions. Figure 1 shows the different villages covered by the producer surveys in the target areas. As the site selection was deliberate, the results can only be considered to reflect the situation in the specific villages, although some aspects are likely to be common to other sites in southern Mali.

In each village, producers of fonio and Bambara groundnut were invited to participate in focus group discussions. Local farmer leaders supported the selection of participants, which aimed to include six to ten knowledgeable producers of each crop with an equal gender split. Final participation depended on producers' availability and interest. Across the eight sites, a total 179 producers participated in the discussions, of which 56% were women. The focus groups were held in May 2017 and led by a researcher from Bioversity International, with support from translators and IER staff. Following a semi-structured interview approach, the discussions explored how fonio and Bambara groundnut were considered by farmers, cultivation practices, variety characteristics, and input requirements, especially the labor intensity of the different steps in the production process. Strengths and weaknesses of growing the crops were assessed, together with their general contributions to local livelihoods. Following the focus groups, a subset of the participants was invited to take part in individual surveys that were designed to gain further insights into the production and consumption patterns of the target crops. A total of 25 fonio producers (16% women) and 31 Bambara groundnut producers (29% women) were interviewed individually. The target of an even gender split could not be achieved with the individual surveys because fonio was almost exclusively farmed by men in the focal sites, and women were less free to attend meetings with researchers and NGO workers, requiring permission from their heads of household.

### 2.2.3. Consumer Surveys

The focus groups and individual surveys with producers in 2017 included questions regarding preferences and perceptions with respect to consuming fonio and Bambara groundnut. This approach was made in acknowledging the villagers as both producers and consumers of the crops. In addition, during the 2018 data collection carried out in Bamako, we surveyed 72 consumers of fonio, aiming at a balanced sampling across the six communes and the focus fonio products. We randomly selected three neighborhoods in each commune to survey and six consumers within each neighborhood.

### 2.2.4. Analysis

The analysis of the information collected in the surveys focused on identifying common and unique issues raised in interviews within and across levels of value chain actors following a qualitative approach. Summary statistics (means and percentages) were calculated to highlight the prevalence of themes raised by the respondents and to establish average conditions in the value chains as a point of reference. All calculations were made using Microsoft Excel and SPSS.

## 3. Results

### 3.1. Trader Surveys

### 3.1.1. Fonio and Bambara Groundnut Products Traded

Six fonio products were found to be traded in the surveyed markets in 2017: Paddy fonio, whitened fonio, washed and dried fonio, precooked fonio, and djouka fonio. Table 2 provides a summary of each product. In 2018, a total of 43 brands of fonio products were identified in the markets, 33 of which were producing both djouka and precooked fonio, seven of which produced fonio djouka, and three of which produced precooked fonio (Table S1). No brands as such were identified for paddy fonio because the product involves much less processing, it is not packaged, and it is typically sold by the weight in neighborhood markets. Three brands were particularly represented for precooked and djouka fonio: '*Djouka précuit*', '*fonio précuit*', and '*jumeaux prestation*'. Although these brands were found in all types of outlets, products from the '*djouka précuit*' and '*fonio précuit*' brands were mostly found in neighborhood markets, while products from '*jumeaux prestation*' tended to be represented equally in neighborhood markets and grocery stores.

**Table 2.** Fonio and Bambara groundnut products traded.

| Crop/Product | Description | Level of Processing |
|---|---|---|
| Fonio | | |
| Paddy | Threshed and winnowed fonio as the first step of fonio processing | Basic |
| Whitened | Hulled fonio that receives extra processing to remove the bran (pericarp and germ). | Basic |
| Washed and dried | Whitened fonio that has received additional washing and is subsequently dried | Intermediary |
| Precooked | Steamed, washed, and dried fonio, subsequently dried and packaged | |
| Djouka | Precooked fonio mixed with steamed and crushed groundnuts and potash, subsequently dried and packaged | |
| Bambara groundnut | | |
| Grains | Dried Bambara groundnut grains (seeds) | Basic |
| Roasted | Roasted Bambara groundnut nuts following harvest season | Basic |
| Boiled | Fresh nuts boiled after harvest | Basic |

As shown in Table 2, a lower diversity of products was marketed for Bambara groundnut than for fonio, which were: Bambara groundnut grains, roasted Bambara groundnut, and boiled Bambara groundnut. All of these products involved little processing. Bambara groundnut grains were sold as dried products that could be preserved for several months after harvest. Roasted Bambara groundnut was processed at home by women, who then resold it to traders. Boiled Bambara groundnut was primarily sold when the pods were fresh.

### 3.1.2. Trader Characteristics

In 2017, eight traders were interviewed regarding paddy fonio, seven regarding washed and dried fonio, 45 regarding whitened fonio, 11 regarding precooked fonio, 12 regarding djouka fonio, 18 regarding Bambara groundnut seeds, 26 regarding roasted Bambara groundnut, and one regarding boiled Bambara groundnut. The survey was completed approximately six months following the harvest season, when boiled Bambara groundnut is not typically available, which limited our opportunity to survey traders about this product. In 2018, an additional 41 traders in Bamako were interviewed regarding whitened fonio, along with 116 for precooked fonio, and 97 for djouka fonio.

The traders included retailers (selling mainly to individual customers in small amounts), semi-wholesalers (selling larger amounts to individual consumers and small businesses), and wholesalers (selling large amounts in bulk, usually to retailers or larger businesses) (Table 3). The majority of fonio traders interviewed in 2017 were retailers (59%), almost a third were semi-wholesalers (30%), and a minority were wholesalers (7%). In 2018, the distribution of retailers, semi-wholesalers, and wholesalers interviewed in Bamako was similar. More wholesalers and semi-wholesalers were interviewed regarding more highly processed products (djouka and precooked fonio), while more retailers were interviewed regarding products with more limited processing (paddy fonio and whitened fonio). Most (91%) of the Bambara groundnut traders surveyed were retailers, and just a handful were semi-wholesalers (9%). No wholesaler of Bambara groundnut products was identified in the survey. Most traders (78%) owned their business.

**Table 3.** Number of traders surveyed by type (and percent women) by crop in the two survey years.

| Year | Crop | Retailers | Semi-Wholesalers | Wholesalers |
|------|------|-----------|------------------|-------------|
| 2017 | Fonio | 49 (53%) | 25 (72%) | 9 (44%) |
| | Bambara groundnut | 41 (85%) | 4 (100%) | 0 (0%) |
| 2018 | Fonio | 208 (78%) | 37 (5%) | 9 (33%) |

A considerable involvement of women in trading of both crops was recorded. This was the case for Bambara groundnut in general, for which 87% of traders interviewed were women. Precooked and djouka fonio especially were marketed largely by women. Many of these fonio products were processed by women themselves, especially in Bamako, where women processing groups were targeted.

### 3.1.3. Trends and Features of Fonio and Bambara Groundnut Trading

All of the traders of fonio and Bambara groundnut interviewed in Bamako in 2017 sourced their product from collectors, who act as intermediaries between producers and traders. The majority of traders in the Ségou and Sikasso regions likewise sourced fonio and Bambara groundnut from collectors (61% of fonio traders; 50% Bambara groundnut). Others in Ségou and Sikasso obtained fonio and Bambara groundnut from local producers (fonio 14%; Bambara groundnut 23%), their own farm (11% fonio; 13% Bambara groundnut), or, in the case of fonio, from a processing group (14%).

The interviews with traders in 2017 revealed that the period of abundance for paddy and whitened fonio runs from September to December. However, traders were able to stock up enough produce to sell all year round. During the period of abundance, paddy fonio was sold on markets at an average price of 225 FCFA (FCFA is the West African CFA Franc with ISO 4217 code XOF, for which the exchange rate in January 2020 was 1 euro for 654 FCFA.) per kg, while whitened fonio was sold at an average price of 441 FCFA per kg. The higher price reflected the heavier amount of processing involved in generating whitened fonio. During the scarcity period, the average selling price of paddy fonio rose by 59% to 358 FCFA per kg, and that of whitened fonio increased by 29% to 569 FCFA per kg. The reported selling prices of second-level processed fonio products during the abundance period were much higher for washed and dried, precooked, and djouka fonio at 786, 1055, and 1379 FCFA per kg, respectively. Interestingly, the prices of these more intensively processed fonio products were

not reported to rise substantially during the scarcity season. The period of abundance for Bambara groundnut also ran from September to December. The average selling price for Bambara groundnut seeds during abundance was 487 FCFA per kg in our sample, while that of roasted Bambara groundnut averaged 637 FCFA per kg, which was consistent with the slightly higher level of processing involved in the latter. During the scarcity period, prices rose by 9% to 533 FCFA per kg for the grains, and by 18% to 752 FCFA per kg for the roasted grains.

The surveys in Bamako in 2018 revealed that the average sales of whitened fonio for traders in the previous year (2017) amounted to 3100 kg, while sales of precooked and djouka fonio amounted to 7103 and 2750 kg, respectively. Some traders were engaged in exporting fonio products, but the volume of sales represented less than 3% of the total volumes traded across all traders interviewed. In terms of destinations, most of the exported fonio went to Senegal and France to cater for diaspora markets. Both countries imported precooked and djouka fonio. Processing companies, which sold their products to surrounding markets in Bamako, processed an average of 1020 and 783 kg of precooked and djouka fonio per week, respectively. These processing groups needed to source substantial volumes of groundnut to produce djouka fonio. Part of the produce from the processing units was sold outside Bamako to other regions of the country.

The main constraints to marketing paddy fonio reported in 2017 were the slow pace of trade (60% of respondents), low capital (20%), and the presence of sand in the product (20%). For whitened fonio, major constraints were the lack of customers (62%), lack of financial/technical means (27%), issues with suppliers (8%), and low capital (3%). For all three second-level processed fonio products (washed and dried, precooked, and djouka), lack of processing material and presence of sand were the most important constraints. These results were consistent with those from the 2018 surveys. In terms of bottlenecks and constraints to marketing, 31% of traders surveyed in Bamako in 2018 reported no constraints. However, 19% of them mentioned the weak demand for products outside festive periods (Christmas, Tabaski, Ramadan), 25% reported marketing problems due to unstable prices, 9% reported quality issues with the fonio they process (for example, the presence of sand), and 7% of traders complained that supply was not always available. Respondents in processing companies listed lack of equipment and capital as major constraints (44% and 24%, respectively) towards greater fonio processing and commercialization. A secondary but nonetheless important constraint for processing units was lack of space to carry out activities for 13% of respondents. Concerning Bambara groundnut, 50% of roasted Bambara groundnut sellers surveyed in 2017 cited low capital as the main constraint related to marketing, followed by lack of buyers/consumers (46%) and high buying price of grains (4%).

*3.2. Producer Surveys*

3.2.1. Production Overview

Focus group participants explained that fonio was cultivated in monoculture. Land preparation started in May, with sowing in June and harvesting in September. After plowing, fonio was broadcast sown. The seed was obtained either from saved seed, purchase in the market, or exchange with another producer, and was not subjected to any prior treatment. Maintenance of the crop involved manual weeding. Harvesting was done by mowing with a sickle in the direction of lodging. During mowing, one hand held a tuft of fonio stalks and the other hand cut it with the help of the sickle a few centimeters from the ground. The stems were gathered into bundles and tied with bark. Most of the tasks involved in cultivation of fonio (soil preparation, sowing, pest control, harvesting, and drying) were carried out by men. Women were involved in weeding, and they had the primary role in processing (winnowing, de-hulling, and removing sand). It was emphasized in both the focus groups and individual surveys that fonio production was very labor intensive. Harvesting and weeding were the most tedious tasks according to 64% and 34% of respondents in the individual surveys, respectively.

The focus groups appreciated the hardiness of fonio, stating that "plots that do not succeed with other crops succeed with fonio", and that it is "not demanding in terms of inputs". At the

same time, availability of fixed inputs (tools, machinery, seeds, rainfall) also came up as a bottleneck in the interviews. Difficulties in accessing improved seeds were raised in four villages (Finkoloni, Somo, Chocolomba, and N'Goutjina), and lack of equipment (especially threshing machines) was mentioned in all eight villages surveyed. In three villages (Kiby, Siramana, and Chicolomba), farmers explained that lack of training was an important barrier towards improved production. Beyond factors hindering cultivation as such, harvest and post-harvest losses were also important factors negatively affecting production. Some losses were caused by climatic conditions (lack of rainfall especially), or by pests, such as striga (*Striga* sp.), and were therefore hard to avoid. However, other losses were the direct outcome of farming patterns. In Chicolomba, farmers explained that fonio and cotton harvests coincided. Because cotton was a key cash crop in the region, its harvest took priority over that of fonio, thus causing fonio losses. However, land availability was not considered a constraint to fonio production. In six out of the eight villages in the study area, farmers mentioned land availability as a factor acting in favor of fonio cultivation, together with availability of working tools and labor.

Bambara groundnut had a similar planting and harvesting schedule to that of fonio. It was cultivated in very small areas. While soil preparation and transportation were men's tasks, shelling, winnowing, and sorting were women's tasks. Furthermore, women tended to be more involved in weeding than men. In contrast to the other sites, in Siramana, Bambara groundnut cultivation was a predominantly woman-led activity. During cultivation, it suffered from important losses due to pest and disease attacks. Post-harvest losses were also frequent, with grains being eaten by insects due to lack of proper storage units. Climate constraints were also named in four villages, especially insufficient rains, and also too much rain during the harvest period. In terms of factors positively affecting adoption and growing, producers mentioned availability of land, seeds, and labor. Seed availability was noted as a strength in three sites in Ségou and one site in Sikasso, while lack of information on improved seeds was named in two sites in the Sikasso region. The results pointed out the importance of Bambara groundnut production and sales for women, as it constituted an independent source of income.

### 3.2.2. Producers' Marketing

Fonio was often produced with the aim of carrying first-step processing within the household. Producers marketing fonio did so via one of three channels. The first one consisted of selling directly to consumers within village boundaries; the second to collectors, who would then sell the product to traders in markets; and the third channel consisted of selling raw produce to hulling mills, typically via a cooperative or farmer group. Such mills could be found in San and Somo, for example. After processing, the mills would sell the transformed fonio to traders and especially wholesalers. Perceptions of the market for fonio were mixed in the different communities. Producers in three communities in Sikasso considered that there was no specific or local market, and there was generally low demand for fonio products. Producers in the Ségou region explained that the market is unstable, sometimes passing from 1000 to 250 FCFA per kg. Washed and dried fonio generally had a better price and market than unprocessed fonio. Some producers mentioned specifically growing improved varieties of fonio that yield bigger grains, as this helped them get a higher price. As mentioned earlier, fonio production was usually taken care of by men, whereas its processing was mainly a women's activity. Participants explained that women specialized in fonio processing rather than production because they were not landowners, and because many tasks involved in its farming were difficult to carry for women.

The marketing of Bambara groundnut was an activity only conducted by women in the study communities. This crop was typically sold within village boundaries in small volumes. Processed at home, it was mostly sold roasted and sometimes boiled. In a few instances, some women in Somo and N'Goutjina explained that they also had informal contacts with traders to sell their produce. Respondents in Siramana, Chicolomba, and Somo considered that there was "no good market", "no market with a good price", and "no specific market" for Bambara groundnut. By contrast, the crop was considered to have a good market by producers in N'Goutjina. Female participants in the focus

groups complained about a lack of processing facilities for Bambara groundnut, pointing to a key gap in the existing value chain. This gap went hand in hand with a general lack of information on the crop regarding its marketing potential. In that respect, the gap between fonio and Bambara groundnut was important.

### 3.3. Consumer Surveys

### 3.3.1. Rural Consumer Perceptions

Consumer perceptions were assessed during the focus groups and individual surveys with producers in the Ségou and Sikasso regions in 2017. Fonio is used to prevent hunger in the family before other cereals are in maturity. Fonio also had ceremonial values, being used in marriage and to make sacrifices. Respondents mentioned that "when you eat fonio, you stay long without being hungry". Digesting fonio was considered easier than digesting rice, and it was recognized to be a good food for diabetics. Fonio was cooked in several forms: *Foyo, tô*, and *lafri* (like *riz gras*). Participants mentioned that their parents and grandparents did not know pre-cooked and djouka fonio, and in Finkoloni, women did not know how to process these forms. For ease of consumption, it was important that grains were clean of impurities, especially sand, and well dried. Participants explained that best practices entailed drying the crop over a day indoors and covering it with a cloth to limit contamination with sand as much as possible.

Similarly to fonio, the possibility of consuming Bambara groundnut as a backup food ('*aliment de secours*') was important. Producers described it as an early crop that helps in difficult times when there is no food. It was also used as a snack that "supplements the food cereals between morning and noon and noon and evening". There was an awareness of the nutritional properties of the crop. For example, participants said it "plays the role of meat in the body" and that "children who eat it do not get sick of malnutrition". Bambara groundnut was made into croquettes, galettes (grilled flatbreads mixed with okra powder), and *tô* (crushed seeds mixed with millet or sorghum flour and water).

### 3.3.2. Urban Consumer Perceptions

In the 2018 Bamako surveys, the vast majority of fonio consumers surveyed were women working as traders on the markets or as housewives. Similarly to the results in the rural areas, respondents praised the popularity of the crop during traditional and religious events. For example, several respondents mentioned that precooked fonio is consumed during the Tabaski festival, while one respondent highlighted consumption of djouka fonio during the Christmas season. Precooked fonio was typically bought in stores, while most whitened and djouka fonio was bought in markets. A recurrent complaint among respondents was that prices for all fonio products were unreasonably high. Their price fluctuated around the year with seasonal availability (precooked fonio was less available during the rainy season, for example), but also due to the presence of traditional festivals, which increased its selling price. Issues related to cooking difficulty and the presence of sand or small stones in all fonio products were also routinely reported by consumers. Among popular fonio dishes were *foyo*, typically preferred by consumers of whitened and precooked fonio, and red djouka, preferred by consumers of djouka fonio. Praised for its nutritional content and the ease with which it is digested, most respondents reported a willingness to consume more fonio should their constraints be lessened.

## 4. Discussion

Our interviews with traders, producers, processors, and consumers point to a series of constraints that limit the consolidation of fonio and Bambara groundnut value chains. Table 4 summarizes the issues for different actors of the value chains. Assessing the constraints along the different stages of a value chain helps understand why these minor crops remain so, despite their nutritional and economic potential [46]. The development of functional value chains is important to consolidate resilient

agricultural alternatives while providing local populations with nutritious foods [47]. Neglected and underutilized species tend to have bottlenecks across their value chains, including agronomic constraints (low yields, poor access to seeds, etc.), commercialization constraints (poor value chain organization, high transaction costs), and weak consumer demand as a result of low awareness, negative perceptions of the crop, or difficult processing, among other factors [46,48]. These bottlenecks can reduce interest in growing and using species, while other crops compete for time, land, and space in local diets and use practices. As Table 4 shows, the value chains of both fonio and Bambara groundnut faced barriers along their value chains that limited their further integration into the Malian food system.

**Table 4.** Summary of issues in the value chains for fonio and Bambara groundnut detected in the surveys with actors at different levels.

| Value Chain Actor | Fonio | Bambara Groundnut |
|---|---|---|
| Traders and processors | • Paddy fonio: Slow trade, low capital, presence of sand<br>• Whitened fonio: Lack of customers, issues with suppliers, low capital<br>• Second-level processed fonio: Lack of processing material, presence of sand<br>• Processing units: Lack of equipment, lack of capital | • Roasted Bambara: Low capital, lack of customers, high buying price for grains |
| Producers | • Difficulties in accessing improved seeds<br>• Lack of machinery, especially threshing machine<br>• Lack of formal farming training<br>• Harvest coincides with cotton<br>• Recurrent pests and diseases<br>• Insufficient rain | • Lack of processing facilities<br>• General lack of promotion and interest in the crop seen as 'petty' trade<br>• Climate constraints (not enough rain, too much at harvest time) |
| Consumers | • High prices in Bamako<br>• Presence of sand and/or stones in the product | • High price of roasted Bambara<br>• Poor quality of transformed product |

*4.1. Overview of Bottlenecks*

The lack of a reliable market stood out as a limitation that demotivated farmers to produce fonio and Bambara groundnut commercially and traders to deal with these crops. Farmers sold their harvests within their villages to local consumers, as well as to collectors and mills, but they complained about a lack of a consistent market and good prices. For traders, low consumer demand stood out as a primary constraint for commercialization of fonio and Bambara groundnut, as captured by observations of a "low pace of trade" and "lack of customers". The consumer interviews confirmed observations of low demand that were raised by traders and farmers. Both crops were used mainly as accessory foods under specific circumstances and were not regular staples of diets year-round. Rural consumers relied on these crops for food security in the lean season because of their early maturation, which is a role that has been well documented for fonio [49]. Bambara groundnut was used as a snack to hold over the appetites of rural consumers between meals. Fonio additionally held important ceremonial roles for both rural and urban consumers, which traders observed to concentrate demand in a few periods. Fonio was generally held in high esteem by consumers, as is common through several areas of West Africa, where it is appreciated as a superior crop or a diversification product in urban areas [15]. However, the quality of the processed products, which can often contain sand, reduced their desirability. High prices were also a barrier for greater use of fonio by consumers. In this sense, our results are consistent with prior research that has stressed the detrimental effects of high prices on sustained consumption in urban Mali [27] and in the West African region [15]. Bambara groundnut did

not hold the same positive esteem as fonio, and it was mainly seen as a backup crop for food security. Subsistence crops can carry a stigma as food of the poor, which poses a barrier to their full integration in food systems, as has been observed, for example, for African leafy vegetables [50], and this may be a factor holding Bambara groundnut from wider integration in the Malian food system.

For fonio, a clear difference emerged between products with low processing (paddy and whitened fonio) and those with more advanced processing and packaging (precooked and djouka fonio). Producers described the market for more advanced fonio products to be more reliable and with better prices compared to that for paddy fonio. This was consistent with our results from the trader surveys, which revealed more stable pricing of advanced fonio products over the year as compared to products with lower levels of processing. Some women in the rural areas were engaged in processing fonio as an income source, but the scale of commercial fonio processing in the villages was not substantial. Producers across the sites complained of missing threshing machines and other processing equipment. Most of the threshing was still done manually by beating the straw, which is a very labor-intensive activity that often yields poor results in terms of product quality [30,51]. A lack of processing equipment has been observed elsewhere as a barrier for fonio commercialization at the levels of producers and processors (e.g., [30] for Togo). Mills in district centers in the Ségou and Sikasso regions were involved in processing fonio, and numerous women's processing groups in Bamako were active in sourcing paddy fonio to process it for sale in grocery stores, supermarkets, and other markets. Lack of capital was a general complaint by processing groups and retailers of fonio. These issues connected back to consumer demand, as the presence of sand in the fonio products can be seen as a side effect of inappropriate technology and low capital. This also translates into complaints regarding the high price of the product, which requires a large investment of time, drudgery, or capital to process.

Lack of capital was an equally important constraint for traders of Bambara groundnut, together with poor product quality. Here too, our results largely reflected findings from other studies: The literature on Bambara groundnut stresses the lack of functional value chains for the crop, resulting in irregular markets (see, for example, [32] for Ghana, or [19] for an African overview). One aspect we saw in our results that was more specific to Bambara groundnut was the lack of systematic promotion of the crop, which hindered its production and processing by farmers. Bambara groundnut is often considered a women's crop [33], and its marketing is confined to the margin of households' main agricultural strategies [31]. In the Ghanaian context, the crop is also found to be grown more by females than males [34]. Because the latter have little interest in expanding production and commercialization of the crop, it remains largely invisible in terms of promotion and advertisement. As such, male farmers and traders are rarely interested in expanding its cultivation and sales. Even though Bambara groundnut is popular as a snack food and was praised for its nutritional properties by producers and consumers, its widespread lack of promotion reduces its visibility. A good example was the absence of Bambara groundnut dishes in restaurants and supermarkets in the study sites, as was anecdotally noted by the research team. Product diversification for Bambara groundnut was also very limited despite the versatility of processing that is possible for this crop, especially by grinding it into flour [52]. Biological issues, such as long cooking time, may still constitute an important problem in this regard. The required energy to transform Bambara groundnut within the household is often too high to make processing a viable activity [31,34].

There were a few agronomic constraints mentioned in the producer interviews that contribute to the limited production of Bambara groundnut and fonio. Low rainfall reduces yields of both crops. In this sense, although fonio and Bambara groundnut are generally hardy with respect to local conditions, they are not immune to climate change. For example, drought was a factor that contributed to abandonment of this crop by producers in savannah areas of Ghana and Nigeria [34,53]. Storage pests were a major issue for producers of Bambara groundnut, which agrees with observations of this crop in northwestern Nigeria [54]. A poor availability of seed—especially seed of improved varieties—was mentioned for both crops in both regions. This complaint is not surprising given that no improved varieties for Bambara groundnut have been released in Mali. By comparison, several

improved varieties of fonio have been released in Mali, but they are not always accessible to farmers, and they tend not to perform reliably better than local varieties [55].

Given the popularity of fonio in the region, its importance in traditional events, and consumers' willingness to consume more of it, the crop can be considered underutilized in the sense that its commercial potential is not realized. The underutilized status of Bambara groundnut is equally clear, if not more. Product transformation is very limited, and roasted seeds are the main product available. Processing is carried out within producers' households, and trade is confined to local areas due to the lack of promotion of the crop. Strengthening the value chains of these crops therefore requires targeted interventions to alleviate these specific bottlenecks. However, both bottlenecks and policy recommendations need to be analyzed with gender lenses, as there are clear gender patterns observed in the cultivation, processing, and marketing of the target crops.

*4.2. Policy Recommendations*

To assess the policy interventions required to mainstream fonio and Bambara groundnut, the framework developed in [56] is a useful benchmark. Concerned with how to improve marketing of marginal crops in a way that directly benefits the poor, the authors argue that three conditions are necessary (though not necessarily sufficient) for successful commercialization of underutilized species. First, demand needs to expand and stay sustained over time. To this end, marketing and promotional campaigns, advertising, and information dissemination are important. Second, efficiency along the value chains needs to be increased to bring consumers a product of quality for a reasonable price. This involves, for example, the organization of farmer groups or cooperatives that help reduce intermediaries' bargaining power, and the development of processing units. Both these interventions can be addressed through local and national policies. They also entail better communication between different actors of the value chain, as well as decent infrastructure and transport networks to minimize transaction costs. Finally, a situation in which production soars and subsequently depletes prices must be avoided, as it would discourage farmers from continuing to produce marginal crops. One possible way to avoid this is to promote quality-rewarding mechanisms; for example, by specifying necessary quality attributes that a product must have in order to be commercialized through certification labels. The development of certification and labeling systems to support the marketing of NUS products by local communities is a policy intervention that has been poorly explored insofar at both the national and international levels. Simple and inexpensive certification mechanisms linked to on-farm conservation of NUS diversity and to resilience and nutritional benefits arising from their greater use should be more decisively taken up by governments and agencies concerned for the wellbeing of local populations [57]. Additional national policy measures that could be explored to strengthen the value chains of NUS in Mali include the introduction of these nutritious foods in public procurement schemes (e.g., school meal programs), as was done recently in Guatemala [58], or the establishment of multi-stakeholder platforms involving all value chain actors, as was done successfully in the case of amaranth in Bolivia [59].

4.2.1. Gender Dynamics in Processing and Trading

Even though the value chains for fonio and Bambara groundnut products are quite distinct, several barriers to improved trading are common to female traders marketing both crops. Future value chain interventions should aim at retaining women in the trade of marginal crops while simultaneously helping them overcome the disproportionate barriers they are facing (access to capital, credit, processing units, etc.).

While efforts should target increased visibility of the crop and the creation of processing units outside the household in the case of Bambara groundnut, interventions for fonio would require instead the strengthening of already existing processing structures. This involves securing availability of credit and provisioning of processing materials to these groups on a perennial basis. Investing in second-level processing technologies is also important for traders because processed products are less subject to price variations than raw or barely processed ones. During our surveys with traders in 2017 in the

Ségou, Sikasso, and Bamako regions, respondents explained that the price of paddy and whitened fonio varied extensively between the abundance and scarcity seasons, whereas prices for washed and dried, precooked, and djouka fonio did not substantially rise in the scarcity season. Therefore, policy interventions focusing on local development should ensure that the necessary technologies for second-level processing are available to women's processing groups who supply traders with fonio (as well as restaurants). Given that traders of processed fonio also tend to be females, this would strengthen women's position along the value chain. Creating well-functioning formal markets is a key condition for the consumption of traditional crops to consolidate and flourish, as shown by [60] in the South African context.

A further possible issue in the case of fonio could arise as the value chain consolidates: Fonio trade is currently managed by women, who also retain the income earned from this trade. A potential threat lies in a gender-reversal phenomenon if fonio consolidates as a cash crop and heads of households, who also own the land, embrace its production as a new source of income at the expense of women and their autonomy. This possibility is less immediate for Bambara groundnut given its more marginal status, but it exists in principle too, as stressed in [33].

### 4.2.2. Visibility and Knowledge of the Crops

At the trader level, lack of customers was often mentioned as a barrier towards greater commercialization. Demand is usually high during traditional events or festivals, but is not sustained throughout the year. Agricultural extension policies to stimulate demand through enhanced knowledge and awareness of the crops' benefits would be a useful first step towards achieving this goal. This is the case for both fonio and Bambara groundnut, but the latter suffers from a lack of status and the social stigma of being an 'inferior food' (a condition common to many underutilized crops), which further hinders its trade; efforts are thus required to mainstream the crop through advertisement campaigns, promotion of its nutritional properties, dissemination of recipes, and providing appropriate incentives for traders to promote it. There is evidence that crop promotion influences producers' adoption. For example, the authors of [61] document the case of adoption of annual legumes (including Bambara groundnut) among maize-growing farmers in central Malawi following promotion campaigns conducted between 1998 and 2004. The issue of visibility for Bambara groundnut is linked to its gender dynamics too, as already explained.

### 4.2.3. Access to Inputs and Machinery

For fonio producers, a key lever to increase production is to ensure a better access to quality seeds at an affordable price and to enhance access to technologies that can facilitate harvest and post-harvest treatment of fonio. In particular, the unavailability of threshing machines proved to be a key bottleneck for producers. Pilot schemes have been launched to test the efficacy of threshing and dehusking machines in West Africa [62]. Efforts in this direction should be pursued to assist producers in the development of modern technologies for fonio.

### 4.2.4. More Resilient Cropping Patterns

Although established cropping patterns are hard to change in the short term, efforts by policymakers in Mali should be made to create more space and opportunities for NUS, such as fonio or Bambara groundnut, and to rebalance crop mixes away from environmentally damaging crops, such as cotton. For instance, currently, in Mali, only cotton, maize, and rice benefit from government subsidies, in spite of the fact that Bambara groundnut is able to enrich the soil by fixing atmospheric nitrogen, and both this crop and fonio can grow on low fertile soils, do not suffer from major insect and disease problems, and can generate revenues for poor farmers in areas where cotton cultivation is no longer feasible.

Land availability was not mentioned as a constraint in the case of Bambara groundnut. Interestingly, producers mentioned that labor availability was not an issue either, contrary to fonio. Given the

availability of land, labor, and improved seeds for Bambara groundnut, it is therefore expected that production could easily expand if the processing and commercial features of the crop were developed at higher levels along the value chain, and the right incentives were set up for producers through an enabling policy environment.

## 5. Conclusions

The value chains of fonio and Bambara groundnut in Mali suffer from important bottlenecks, hindering further commercial integration in the case of fonio and the creation of decent marketing opportunities in the case of Bambara groundnut. Among the main bottlenecks for fonio are the lack of demand outside traditional events, presence of sand in the product, and limited access to capital to expand processing and marketing activities. In the case of Bambara groundnut, limited visibility of the crop, lack of processing units, and lack of demand are the most important constraints. Despite these bottlenecks, both crops are important for local livelihoods and are well suited to the environment in which they are grown, where they can perform better than other more popular crops because of their hardiness to local climate conditions. With increasing awareness of the impact of climate change on agricultural output in Western Africa and its effects on livelihoods, embracing agricultural strategies that push these crops to the forefront is of crucial importance.

**Supplementary Materials:** The following are available online at http://www.mdpi.com/2071-1050/12/11/4766/s1, Table S1: Fonio brands from 2018 Bamako market surveys, Supplementary File 1. Interview guidelines.

**Author Contributions:** Conceptualization, C.M., S.P., G.M., and B.B.; Methodology, C.M., G.M., Y.M., A.S., and S.P.; Validation, A.S., Y.K., A.B.N., and H.C.; Formal Analysis, C.M. and Y.M.; Investigation, C.M., Y.M., A.B.N., and H.C.; Resources, S.P. and A.S.; Data Curation, C.M. and Y.M.; Writing—Original Draft Preparation, B.B. and C.M.; Writing—Review and Editing, G.M., S.P., and A.S.; Visualization, C.M. and G.M.; Supervision, S.P. and A.S.; Project Administration, A.S., G.M., and S.P.; Funding Acquisition, S.P. All authors have read and agreed to the published version of the manuscript.

**Funding:** This work was carried out in the framework of an international effort "Linking Agrobiodiversity Value Chains, Climate Adaptation, and Nutrition: Empowering the Poor to Manage Risk" (Grant No. 2000000978), funded by the European Union and the International Fund for Agricultural Development (IFAD), and the CGIAR Research Programs on Agriculture for Nutrition and Health (A4NH) and Climate Change, Agriculture, and Food Security (CCAFS).

**Acknowledgments:** We are most grateful for the time and active participation of the traders, processors, producers, and consumers surveyed. The support of translators and local facilitators from *Centre d'Appui à l'Autopromotion pour le Développement* (CAAD) and *Aide au Sahel et à l'Enfance Malienne* (ASEM) was essential to the success of the study.

**Conflicts of Interest:** The authors declare no conflict of interest.

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
