# Peer review of "Fonio and Bambara Groundnut Value Chains in Mali: Issues, Needs, and Opportunities for Their Sustainable Promotion"

_sustainability, doi:10.3390/su12114766_

Round 1

Reviewer 1 Report

The paper is in general very useful and I've appreciated in particular the thorough research design.

Editorially, in line 222, it should be "consumer surveys" i think, not "producer".

On the "Results", "3.3.2. Urban Consumer Perceptions" gives no info on bambara consumption; if available, useful to mention such info as well. More broadly, I wonder if anything more could be said about the 'advantages' in terms of climate adaptation of the 2 target crops, given the centrality of Climate Change in the Intro and Conclusions of this paper? Any evidence, or perceptions in relation to climate adaptation that possibly emerged in particular from the producers surveys? 

On the "Discussion", maybe the Authors could elaborate more on the kind of "bias" in the Malian food system that works in favour of more commercial crops (hence 'against' the 2 target crops)? This is briefly mentioned ["Because cotton is a key cash crop in the region, its harvest takes priority over that of fonio, thus causing fonio losses" (p.11); and in "4.2.4. Long Term Cropping Patterns"] ...but the implication of subsidies, or other policy incentives, for cotton and other commercial crops, could perhaps be mentioned?

Finally, in "4.2. Policy Recommendations", Authors could:

  • add 1 or 2 sentences to explain more on this idea of "labels" (for sustainability) (line 488);
  • refer to additional measures that could support these 2 value chains, eg.: 1) public procurement schemes (to source more fonio/bambara for schools/hospitals) as incentive for stronger market interest in these chains?  2) Support a specific Multi-Stakeholder Platform in support of fonio/bambara; given Authors mention importance of linking better all the VC actors, this could build trust, coordinate action on production, processing, distribution, consumption and monitor development impact. 3) launch a "Mali NUS Strategy" by the government and other stakeholders (that would start from providing better extension services to farmers to produce fonio and bambara)

Author Response

1st reviewer

Open Review

English language and style

( ) Extensive editing of English language and style required 
( ) Moderate English changes required 
(x) English language and style are fine/minor spell check required 
( ) I don't feel qualified to judge about the English language and style 

Yes

Can be improved

Must be improved

Not applicable

Does the introduction provide sufficient background and include all relevant references?

(x)

( )

( )

( )

Is the research design appropriate?

(x)

( )

( )

( )

Are the methods adequately described?

(x)

( )

( )

( )

Are the results clearly presented?

(x)

( )

( )

( )

Are the conclusions supported by the results?

(x)

( )

( )

( )

Comments and Suggestions for Authors

The paper is in general very useful and I've appreciated in particular the thorough research design.

Editorially, in line 222, it should be "consumer surveys" i think, not "producer".

Response: We do in fact mean the producer surveys and we understand that this is a bit confusing so we have clarified by adding a sentence explaining that “This approach was made in acknowledging the villagers as both producers and consumers of the crops”.

On the "Results", "3.3.2. Urban Consumer Perceptions" gives no info on bambara consumption; if available, useful to mention such info as well. More broadly, I wonder if anything more could be said about the 'advantages' in terms of climate adaptation of the 2 target crops, given the centrality of Climate Change in the Intro and Conclusions of this paper? Any evidence, or perceptions in relation to climate adaptation that possibly emerged in particular from the producers surveys? 

Response: statements have been added to better capture the climate change resilience of target crops (see lines 44, 441, 660 and tab.4)

On the "Discussion", maybe the Authors could elaborate more on the kind of "bias" in the Malian food system that works in favour of more commercial crops (hence 'against' the 2 target crops)? This is briefly mentioned ["Because cotton is a key cash crop in the region, its harvest takes priority over that of fonio, thus causing fonio losses" (p.11); and in "4.2.4. Long Term Cropping Patterns"] ...but the implication of subsidies, or other policy incentives, for cotton and other commercial crops, could perhaps be mentioned?

Response: this has been improved (see lines 768-772)

Finally, in "4.2. Policy Recommendations", Authors could:

  • add 1 or 2 sentences to explain more on this idea of "labels" (for sustainability) (line 488);

Response: this has been done at line 695.

  • refer to additional measures that could support these 2 value chains, eg.: 1) public procurement schemes (to source more fonio/bambara for schools/hospitals) as incentive for stronger market interest in these chains?  2) Support a specific Multi-Stakeholder Platform in support of fonio/bambara; given Authors mention importance of linking better all the VC actors, this could build trust, coordinate action on production, processing, distribution, consumption and monitor development impact. 3) launch a "Mali NUS Strategy" by the government and other stakeholders (that would start from providing better extension services to farmers to produce fonio and bambara)

Response: this has been done in lines 701-705.

Submission Date

22 April 2020

Date of this review

12 May 2020 12:59:58

Reviewer 2 Report

The manuscript by Authors are an interesting and relevant appraisal. There is however some aspects that needs to be addressed in the revised copy. The Comments has been marked on the body of the manuscript (File attached). Beside those, the authors have to follow the comments too as listed below

  1. Introduction is very vast, authors needs to provide a concise and crispy introduction only.
  2. Please add the section of statistics analysis in material and methods along with software used for the statistical analysis.
  3. Conclusion is very vast, authors needs to provide a concise and crispy conclusion in 4 to 5 lines only. Conclusion is the concluding remarks of the present study. Do not quote any reference in the conclusion.
  4. Statistics needs to be added table 1. The significance difference needs to mention with the help of superscripts.
  5. No new reference of 2019 and 2020 are added. Authors are requested to add some more references of 2019 and 2020.
  6. The manuscript needs to be proof read for typographical errors.

Author Response

2nd reviewer

Open Review

English language and style

( ) Extensive editing of English language and style required 
(x) Moderate English changes required 
( ) English language and style are fine/minor spell check required 
( ) I don't feel qualified to judge about the English language and style 

Yes

Can be improved

Must be improved

Not applicable

sufficient background and include all relevant references?

( )

(x)

( )

( )

Is the research design appropriate?

( )

(x)

( )

( )

Are the methods adequately described?

( )

(x)

( )

( )

Are the results clearly presented?

(x)

( )

( )

( )

Are the conclusions supported by the results?

( )

(x)

( )

( )

Comments and Suggestions for Authors

The manuscript by Authors are an interesting and relevant appraisal. There is however some aspects that needs to be addressed in the revised copy. The Comments has been marked on the body of the manuscript (File attached). Beside those, the authors have to follow the comments too as listed below

  1. Introduction is very vast, authors needs to provide a concise and crispy introduction only.

Response: this has been done and we believe the introduction reads well at this point.

  1. Please add the section of statistics analysis in material and methods along with software used for the statistical analysis.

Response: We added an explanation of the analysis and the software used in section 2.2.4.

  1. Conclusion is very vast, authors needs to provide a concise and crispy conclusion in 4 to 5 lines only. Conclusion is the concluding remarks of the present study. Do not quote any reference in the conclusion.

Response: this has been done as requested.

  1. Statistics needs to be added table 1. The significance difference needs to mention with the help of superscripts.

Response: we have not made any statistical comparisons. We have only prepared summary statistics as the analysis has primarily been qualitative. We added a better explanation about the calculations made and presented in the text.

  1. No new reference of 2019 and 2020 are added. Authors are requested to add some more references of 2019 and 2020.

Response: We wish to note that there are already several references from 2019 (e.g. Halimi et al. 2019; Mayes et al. 20219). We added though some additional recent ones (e.g. Mabhaudi et al. 2019; Gaetani et al. 2020).

  1. The manuscript needs to be proof read for typographical errors  

Response:  We have done a careful proof read of the document and made several adjustments to the language to improve grammar and syntax.

Submission Date

22 April 2020

Date of this review

29 Apr 2020 07:58:10

Reviewer 3 Report

Dear Authors,

Thank you for an interesting and highly relevant paper, providing a thorough investigation on the value chains of underutilized crops in Mali. In my review, I have raised several issues, which you will find in the following.

Substantial issues

Aim of the study:

  • While the content of the study is very interesting and relevant, it is not entirely clear what the specific aim / research question of the paper is. Is it investigating why fonio and Bambara are unterutilized? It is investigating what their contribution can be to sustainable agricultural practices and food security? Is it identifying constraints in the respective value chains?
  • The central interest of the paper should be communicated more clearly. The sentence in the abstract "...we investigate issues in their production, commercialization and consumption..." (ll.17-18) illustrates this imprecision.

Section 1 (Introduction):

  • Related to this, the general relevance of the topic should be explained better in the introduction. The last sentences of the conclusion (ll.583-595) actually provide something in this regard. This, as it looks to me, should be in the beginning of the introduction, as it shows the relevance of the article's focus in the wider context.
  • Hence, I would recommend an approach that first outlines the general picture (challenges posed by climate change and the need to sustainable agricultural practices) before introducing NUS as a possible solution and then zooming in on the article's focus on fonio and Bambara.
  • Moreover, the consequences of climate change for West African (and specifically Malian) agriculture should be pointed out. Currently, this is only hinted at in section 2.1 (ll.116-117). What specific challenges does climate change pose to agriculture in the region (citing empirical evidence) and why are NUS such as fonio and Bambara a possible remedy? Moreover, an outlook on / summary of the results is missing in the introduction.

Sample selection and possible bias (referring to Section 2):

  • The criteria employed in the sample selection should be elaborated on in more detail. Convenience sampling is mentioned, but can anything more be said about selection criteria of those interviewed?
  • This also relates e.g. to the descriptive statistics given in Section 3.1.2: were the different groups of traders sampled purposefully? Importantly, the authors should reflect on whether there might be a bias in the results due to the sample selection process.
  • Moreover, it would be helpful to provide a little more background on the organizations the researchers who conducted the survey are affiliated to. Might there have been specific expectations on the part of the respondents biasing responses in one or another direction (cf. l.202 "villages that held good rapportand trust with the researchers' affiliated organizations", while this is very much understandable and makes sense, it should be reflected upon whether this has implications for the data).

Section 4 (Discussion and policy recommendations):

  • Table 4 is very helpful, however, it would be important to have some indication on the issues' relative importance in the table. There is an imbalance between the discussion of the results and the policy implications.
  • Information on the popularity of the crops in the discussion (ll.458-464) should be provided much earlier, ideally in the introduction.
  • In general, the structure of the discussion section could be improved, by summarizing the results in a more structured way and highlighting their implications.
  • Policy recommendations should then follow naturally from these implications. Currently, this is not entirely the case. The policy recommendations are quite lengthy. Parts of what is presented there should rather be moved to the discussion (e.g. ll.508-510; ll.526-528; much of section 4.2.4), so the policy recommendations are "short and sweet" and directly and intuitively emerge from the discussion of the results.
  • The recommendations given in Section 4.2. (ll.475-489) are quite general and widely applicable. In my view, there should be a better connection between the recommendations and the results.
  • It is not clear to me what the policy recommendation is in 4.2.4.
  • It might be useful for the reader if the authors mentioned whom each policy recommendation is adressed to (e.g. national policy-makers, local policy-makers, agricultural extension).

Section 5:

  • Relating back to the clarity on the aim of the study mention in the first comment above, I would encourage the authors to highlight in a more explicit way what the core novelty and additional to exitisting literuature of their study is.

Minor issues

  1. The first sentence of the abstract leaves implicit questions open:
    l.12: "more sustainable development agendas" --> more than what?
    l.12: "cannot [be, seems to be missing] procrastinated any further" --> who has procrastinated in this regard?
    Perhaps rephrase the sentence.
  2. l.34: "persisting" --> is "existing" meant here?
  3. l.103: Better: "Material and Methods"
  4. Section 2.2.1: The fact that convenience sampling was employed (ll.180-181) should be the first point in the description of the sampling procedures (l.171).
  5. Could maps a and b in Figure 1 perhaps be harmonized in terms of their layout?
  6. Sections 2.2.2 and 2.2.3 have the same headline.
  7. Interview and focus group guides should be provided in the appendix.
  8. I recommend splitting Table 2 into one for fonio and one for Bambara.
  9. ll.452-453: What is meant by "systematic lack of promotion"? Perhaps rephrase to "lack of systematic promotion". Same issue appears a few lines further down as well.
  10. The information on the value chains in ll.561-563 (exports) should be providedin the introduction, not in the conclusion.

Author Response

3rd reviewer

Open Review

English language and style

( ) Extensive editing of English language and style required 
( ) Moderate English changes required 
(x) English language and style are fine/minor spell check required 
( ) I don't feel qualified to judge about the English language and style 

Yes

Can be improved

Must be improved

Not applicable

sufficient background and include all relevant references?

( )

( )

(x)

( )

Is the research design appropriate?

(x)

( )

( )

( )

Are the methods adequately described?

( )

( )

(x)

( )

Are the results clearly presented?

( )

(x)

( )

( )

Are the conclusions supported by the results?

( )

(x)

( )

( )

Comments and Suggestions for Authors

Dear Authors,

Thank you for an interesting and highly relevant paper, providing a thorough investigation on the value chains of underutilized crops in Mali. In my review, I have raised several issues, which you will find in the following.

Substantial issues

Aim of the study:

  • While the content of the study is very interesting and relevant, it is not entirely clear what the specific aim / research question of the paper is. Is it investigating why fonio and Bambara are unterutilized? It is investigating what their contribution can be to sustainable agricultural practices and food security? Is it identifying constraints in the respective value chains?
  • The central interest of the paper should be communicated more clearly. The sentence in the abstract "...we investigate issues in their production, commercialization and consumption..." (ll.17-18) illustrates this imprecision.

Section 1 (Introduction):

  • Related to this, the general relevance of the topic should be explained better in the introduction. The last sentences of the conclusion (ll.583-595) actually provide something in this regard. This, as it looks to me, should be in the beginning of the introduction, as it shows the relevance of the article's focus in the wider context.
  • Hence, I would recommend an approach that first outlines the general picture (challenges posed by climate change and the need to sustainable agricultural practices) before introducing NUS as a possible solution and then zooming in on the article's focus on fonio and Bambara.

Sample selection and possible bias (referring to Section 2):

  • The criteria employed in the sample selection should be elaborated on in more detail. Convenience sampling is mentioned, but can anything more be said about selection criteria of those interviewed?

Response: We have explained the sampling protocol as thoroughly as possible. The main criteria for selection of traders was that they were trading the products of interest. Typically there were very few traders of these products and we generally spoke to the only people that were trading them.

  • This also relates e.g. to the descriptive statistics given in Section 3.1.2: were the different groups of traders sampled purposefully? Importantly, the authors should reflect on whether there might be a bias in the results due to the sample selection process.

Response: We do see that some unconscious bias may be present in the sampling and we explained in the methods section that  “the frequency and characteristics of the surveyed traders roughly reflected upon the traders’ and products’ presence in the markets. However, because the sampling was convenience-based, it cannot be considered a strictly representative sample.”

  • Moreover, it would be helpful to provide a little more background on the organizations the researchers who conducted the survey are affiliated to. Might there have been specific expectations on the part of the respondents biasing responses in one or another direction (cf. l.202 "villages that held good rapport and trust with the researchers' affiliated organizations", while this is very much understandable and makes sense, it should be reflected upon whether this has implications for the data).

Response: We have added some clarification that “as the site selection was deliberate, the results can only be considered to reflect the situation in the specific villages, although some aspects are likely to be common to other sites in southern Mali” (lines 259-261).

Section 4 (Discussion and policy recommendations):

  • Table 4 is very helpful, however, it would be important to have some indication on the issues' relative importance in the table. There is an imbalance between the discussion of the results and the policy implications.

Response: We added a description of the approach for the analysis and developed the description of bottlenecks section in the discussion, which has improved the clarity and balance in the paper.

  • Information on the popularity of the crops in the discussion (ll.458-464) should be provided much earlier, ideally in the introduction.

Response: this was done by improving the introduction in several parts

  • In general, the structure of the discussion section could be improved, by summarizing the results in a more structured way and highlighting their implications.

Response: We developed the description of bottlenecks section in the discussion, focusing the paragraphs more clearly around specific topics.

  • Policy recommendations should then follow naturally from these implications. Currently, this is not entirely the case. The policy recommendations are quite lengthy. Parts of what is presented there should rather be moved to the discussion (e.g. ll.508-510; ll.526-528; much of section 4.2.4), so the policy recommendations are "short and sweet" and directly and intuitively emerge from the discussion of the results.

Response: this has been done by shortening recommendations and make them more specific and better targeted within Section 4.2

  • The recommendations given in Section 4.2. (ll.475-489) are quite general and widely applicable. In my view, there should be a better connection between the recommendations and the results.

Response: see previous answer

  • It is not clear to me what the policy recommendation is in 4.2.4.

Response: this has been better explained  

  • It might be useful for the reader if the authors mentioned whom each policy recommendation is adressed to (e.g. national policy-makers, local policy-makers, agricultural extension).

Response: this has been taken care within Section 4.2

Section 5:

Relating back to the clarity on the aim of the study mention in the first comment above, I would encourage the authors to highlight in a more explicit way what the core novelty and additional to exitisting literuature of their study is.

Response: We have pointed to the novelty of the study in the introduction lines 125-130.

Minor issues

  1. The first sentence of the abstract leaves implicit questions open:
    l.12: "more sustainable development agendas" --> more than what?
    l.12: "cannot [be, seems to be missing] procrastinated any further" --> who has procrastinated in this regard? 
    Perhaps rephrase the sentence. done
  2. l.34: "persisting" --> is "existing" meant here? done
  3. l.103: Better: "Material and Methods" done
  4. Section 2.2.1: The fact that convenience sampling was employed (ll.180-181) should be the first point in the description of the sampling procedures (l.171). done
  5. Could maps a and b in Figure 1 perhaps be harmonized in terms of their layout? Response: The figure has been adjusted to provide a more harmonized layout.
  6. Sections 2.2.2 and 2.2.3 have the same headline. done
  7. Interview and focus group guides should be provided in the appendix. Response: We have added the interview guides in supplementary materials.
  8. I recommend splitting Table 2 into one for fonio and one for Bambara. done
  9. ll.452-453: What is meant by "systematic lack of promotion"? Perhaps rephrase to "lack of systematic promotion". Same issue appears a few lines further down as well. done
  10. The information on the value chains in ll.561-563 (exports) should be providedin the introduction, not in the conclusion. done

Submission Date

22 April 2020

Date of this review

02 May 2020 17:54:06